# Poor Alignment and Steerability of Large Language Models: Evidence Using 30,000 College Admissions Essays

## Abstract

People are increasingly using technologies equipped with large language models (LLM) to write texts for formal communication, which raises two important questions at the intersection of technology and society: Who do LLMs write like (model alignment); and can LLMs be prompted to change who they write like (model steerability). We investigate these questions in the high-stakes context of undergraduate admissions at a selective university by comparing lexical and sentence variation between essays written by 30,000 applicants to two types of LLM-generated essays: one prompted with only the essay question used by the human applicants; and another with additional demographic information about each applicant. We consistently find that both types of LLM-generated essays are linguistically distinct from human-authored essays, regardless of the specific model and analytical approach. Further, prompting a specific sociodemographic identity is remarkably ineffective in aligning the model with the linguistic patterns observed in human writing from this identity group. This holds along the key dimensions of sex, race, first-generation status, and geographic location. The demographically prompted and unprompted synthetic texts were also more similar to each other than to the human text, meaning that prompting did not alleviate homogenization. These issues of model alignment and steerability in current LLMs raise concerns about the use of LLMs in high-stakes contexts.

## 1 Introduction

LLMs have been adopted in earnest as a tool for formal communication, such as academic and professional writing. College application essays, in which prospective students describe themselves and their background in response to a provided essay prompt, are an important part of one of the most high-stakes decision-making processes in US education. Yet the potential consequences of students using LLMs to write these essays are not well understood. One concern is that LLMs, despite their accessibility and vast capabilities, may not be equipped to preserve the individuality of each student's writing. These models tend to produce generic, homogenized outputs Liang et al. (2024); Chakrabarty et al. (2024), which could lead students to unknowingly conform to a more standardized and generic way of expressing themselves at a moment when authenticity and highly individualistic narrative writing is strongly preferred Huang (2024); Beck & Godley (2023). Studies have also shown consequential bias in LLMs, such as their tendency to flatten and even penalize the identities of demographic groups Wang et al. (2025); Hofmann et al. (2024) or reveal in-group favoritism in conversations with humans Hu et al. (2025) through the text they generate. This trend poses a challenge from the perspective of evaluators. If more students submit essays influenced by LLMs (or even written entirely by LLMs), the uniqueness of individual applications may start to disappear and they may induce certain biases in their word choices and writing styles that could reflect algorithmic monoculture Kleinberg & Raghavan (2021); Koch & Peterson (2024). Beyond the particularities of college admissions, we analyze comparisons between LLM and human writing to articulate new insights about model alignment and steerability as they pertain to important social processes.

This paper is guided by two critical questions at the intersection of technology and society: (1) Who do LLMs write like (model alignment)?; and (2) Can LLMs be prompted to change what they write like (model steerability)? Alignment is generally understood as the coordination between AI and human tendencies, preferences, and values Kirk et al. (2024). Steerability is generally understood as the degree to which models vary in their responses to different prompting strategies Miehling et al. (2024). Here, we examine alignment and steerability from a more grounded perspective by pairing LLM outputs with actual human writers. We investigate these questions in the high-stakes context of undergraduate admission at a highly selective university by comparing lexical variation between essays written by 30,000 applicants before the release of ChatGPT (by far the most popular LLM interface Zhang et al.) during the 2019-2020 through the 2022-2023 academic years[1]. We compared three types of college application essays: human-written essays, LLM-generated essays simply prompted to respond to the essay question, and LLM-generated essays with additional identity prompting (e.g., "I am a female. I am Asian. I live in Salinas, CA. My parents have college degrees."; we refer to these as identity-prompted essays hereafter). The LLM-generated essays were paired with essays written by actual college applicants with a matching essay question and applicant demographics. Past analyses have shown that admissions essays contain strong demographic imprints from their authors, making them an ideal source of data for our analyses Alvero et al. (2020; 2021). In this study, we examine how closely LLM-generated essays resemble human-written ones and whether identity prompting affects this resemblance across demographic groups.

## 2 Methods

### 2.1 Synthetic essay generation

We employed eight LLMs from four distinct providers to examine variations in college application essays and their susceptibility to identity-based prompting. We randomly sampled 30,000 applicants from applications submitted over a four-year period to the case institution and analyzed their Common App essays. We removed essays shorter than 250 words for a final dataset of 29,232 essays and applicants. Synthetic essays were generated by prompting each LLM using the original Common Application essay questions (see Appendix A.1). For each applicant, we created two essay variants: one incorporating prompts reflecting the applicant's actual identity and another devoid of any identity-related prompts. We chose GPT-4o, GPT-4o-mini OpenAI (2024; 2023), Llama 3 8B, Llama 3.1 70B Dubey et al. (2024), Claude 3.5 Haiku, Claude 3.5 Sonnet Anthropic (2024), Mistral Nemo and Mistral Large Jiang et al. (2023).

The prompted identities included binary gender, first-generation status, race/ethnicity, and the location of the applicant's high school. The location served as a proxy for where a person lives. The essays were generated to be no longer than 650 words. Table 1 describes the full prompt to generate synthetic essays. We conducted several parameter tests to arrive at a set of parameter values for each model that produced coherent essays without strange words (see Appendix A.2 for details).

### 2.2 Preprocessing

In the preprocessing stage, we first filtered out essays that did not meet the maximum word count requirement of 650 words, ensuring that only substantial essays were included in our analysis. Additionally, we excluded essays that provided only a wrapper without any actual content, such as those consisting solely of introductory statements (e.g. "Certainly! Here is the essay."), or closing statements (e.g. "I hope this will help you!"). The wrappers we identified in this preprocessing step are presented in Appendix A.3. Some responses included refusal statements (e.g., "I'm not a human, so I can't write an essay for you."),

---

[1]ChatGPT, which helped popularize generative models with the public, was released November 30, 2022, but essays are typically written beforehand as many application deadlines begin November 1st and end on January 1st, depending on the school. At the time, access to ChatGPT was limited, making it highly unlikely that any of these essays were generated or co-written with LLM assistance.

Table 1: **Prompting to generate synthetic essays**

| System Prompt |
|---|
| You are **[Platform Name (e.g. ChatGPT)]**, a large language model trained by **[Corporate Name (e.g. Open AI)]**, based on the **[Model name (e.g. GPT4o)]** architecture. |
| **User Prompt with Identity** |
| I am a high school student applying to the [case institution]'s College of Engineering. Here is a little bit more information about myself. |
| I am a **[Sex]**. I am **[Race]**. I live in **[Location]**. **[My parents have college degrees/My parents did not complete a college degree.]**. |
| I have to write an essay as part of my application. The essay must be longer than 250 words but no more than 650 words. Below are the instructions for writing the essay and the specific prompt I need to respond to. Write an essay based on the given instructions and prompt. |
| Instructions: "The essay demonstrates your ability to write clearly and concisely on a selected topic and helps you distinguish yourself in your own voice. What do you want the readers of your application to know about you apart from courses, grades, and test scores? Choose the option that best helps you answer that question and write an essay of no more than 650 words, using the prompt to inspire and structure your response. Remember: 650 words is your limit, not your goal. Use the full range if you need it, but don't feel obligated to do so. (The application won't accept a response shorter than 250 words.)" |
| Prompt: "**[Essay question choice]**" |
| **User Prompt without Identity** |
| I am a high school student applying to the [case institution]'s College of Engineering. Here is a little bit more information about myself. |
| I have to write an essay as part of my application. The essay must be longer than 250 words but no more than 650 words. Below are the instructions for writing the essay and the specific prompt I need to respond to. Write an essay based on the given instructions and prompt. |
| Instructions: "The essay demonstrates your ability to write clearly and concisely on a selected topic and helps you distinguish yourself in your own voice. What do you want the readers of your application to know about you apart from courses, grades, and test scores? Choose the option that best helps you answer that question and write an essay of no more than 650 words, using the prompt to inspire and structure your response. Remember: 650 words is your limit, not your goal. Use the full range if you need it, but don't feel obligated to do so. (The application won't accept a response shorter than 250 words.)" |
| Prompt: "**[Essay question choice]**" |

which we also excluded. We observed that Llama 3 8B, in particular, disproportionately returned such refusal responses for the white male group; thus, we excluded the model in several analyses to not bias our results (see Appendix A.5). After this preprocessing step, we retained 29,232 essays for each type of authorship (i.e., 87,696 essays in total, including the human-written essays and the two kinds of LLM-generated essays).

### 2.2.1 Sentence Encoding

To transform the essays into numerical representations, we used the T5 transformer Raffel et al. (2023), specifically the 'tf-base' variant available in the Hugging Face Transformers library. he essays were preprocessed and tokenized with a maximum length parameter of 1024 to handle longer inputs, ensuring sequences exceeding this length were truncated accordingly. Padding was applied to match the longest sequence within the dataset. Following tokenization, the essays were processed using the encoder component of the T5 model, resulting in hidden-state embeddings for each token, specifically utilizing the final hidden states. Given an input sequence $\mathbf{x} = \{x_1, x_2, ..., x_T\}$, the encoder generates contextualized embeddings $\mathbf{H} = \{h_1, h_2, \ldots, h_T\}$, where each embedding $h_t \in \mathbb{R}^d$, and $d$ denotes the hidden size of the model (768 in this study). These contextualized embeddings represent the semantic meaning of each token relative to its surrounding context, differing fundamentally from static embeddings by adapting dynamically to the broader sentence structure and semantics. This adaptability allows the embeddings to capture nuanced contextual information, making them particularly effective for downstream tasks such as text similarity analysis and classification.

After obtaining hidden states from the model, mean pooling was applied to aggregate token-level embeddings into a single vector representing the semantic essence of the entire sequence. The mean pooling calculation considered only valid tokens, identified by an attention mask $\mathbf{m} = \{m_1, m_2, ..., m_T\}$, where $m_t \in \{0, 1\}$ denotes the validity of a token (1 for valid tokens, 0 for padding).

The mean-pooled sentence embedding $s$ is calculated as:

$$s = \frac{\Sigma_{t=1}^{T} m_t h_t}{\Sigma_{t=1}^{T} m_t}$$

This approach ensures that padding tokens do not influence the sentence representation, resulting in an embedding $s \in \mathbb{R}$ that effectively captures the average semantic meaning of all valid tokens.

After applying mean pooling, we convert the vectors into unit vectors by adding L2 normalization. This step mitigates the impact of vector magnitude differences and ensures that subsequent analysis focuses on the directional properties of the embeddings. It is particularly important when measuring distances between two vectors using cosine similarity.

The L2-normalized embedding $s_{norm}$ is computed as:

$$\mathbf{s}_{\text{norm}} = \frac{\mathbf{s}}{\|\mathbf{s}\|}$$

where,

$$\|\mathbf{s}\| = \sqrt{\sum_{i=1}^{d} s_i^2}$$

This normalization step transforms $s_{norm}$ into a vector with an Euclidean norm of 1.

### 2.3 Principal Components Analysis

With the normalized embeddings from the T5 transformer, we explored the structure of the embeddings in a lower-dimensional space. We applied Principal Components Analysis (PCA) Jolliffe (2002) to identify initial clusters within the high-dimensional encoded embeddings. We extracted the first two components to explore the semantic content of the three essay types. These components provided a low-dimensional representation that facilitates an intuitive understanding of the distribution and separability of the essay types in the embedding space.

### 2.4 Pairwise similarity

To evaluate the extent to which LLM-generated essays resemble human writing, we computed the pairwise cosine similarity for different pairs of essays using their corresponding embeddings. Each human applicant was paired with two types of LLM-generated essays: an LLM default essay and an LLM ID-prompted essay.

To establish a baseline for determining the resemblance between LLM-generated and human-authored essays, we employed stratified random sampling among the human essays to create human-to-human pairs. This baseline reflects the typical similarity between essays written by different human authors. Subgroups for human-to-human pairs were stratified based on demographic and contextual factors, specifically essay prompt choice, sex, first-generation college status, race, and U.S. state (or country for international applicants). This ensures a representative comparison across diverse populations. We then calculated pairwise cosine similarity for the following four types of essay pairs:

- **Human vs. Human**: Baseline comparison representing the degree of similarity between essays written by different human authors.

- **Human vs. LLM**: This pair measures how similar LLM-generated essays (without identity prompting) are to human-authored essays.

- **Human vs. LLM ID-Prompted**: This pair captures the similarity between human essays and LLM essays generated using applicant-specific identity prompts.

- **LLM vs. LLM ID-Prompted**: This pair evaluates how similar LLM essays generated with and without identity prompting are to each other.

The pairwise cosine similarity between embeddings $u$ and $v$ is calculated as:

$$\text{Cosine Similarity}(u, v) = \frac{u \cdot v}{\|u\| \|v\|}$$

where $u \cdot v$ denotes the dot product of the vectors, and $\|u\|$ and $\|v\|$ are their respective magnitudes.

For each type of pair, similarity scores were aggregated across demographic categories such as sex, first-generation college status, and race to identify trends and ensure balanced comparisons.

Ultimately, this analysis allows us to examine whether LLM-generated essays closely resemble human-authored essays (both individually and in aggregate) and whether LLM-generated essays are more similar to each other than to human essays. By comparing the similarity of human-to-human pairs with the other pair types, we assess whether LLM essays mimic human writing patterns or form a distinct stylistic cluster. The results are elaborated in Figure 2. The underlying demographic breakdown is also described in Figure 4.

## 2.5 Author classification

We analyzed the distinction between human-authored and LLM-generated essays through a classification approach. We used a Bag-of-Words approach in addition to the sentence encoding generated from Section 2.2.1. We applied text vectorization using the TF-IDF vectorizer. For this encoding, we applied an additional preprocessing step that involved text cleaning, tokenization, stopwords removal, and lemmatization. Specifically, we first removed numbers, non-ASCII characters, and excessive spaces to ensure a clean textual input. Next, we stripped punctuation and converted all text to lowercase. We then tokenized the text into individual words and applied lemmatization, using part-of-speech tagging to ensure accurate word transformations. Finally, we filtered out stopwords and retained only meaningful words with a length greater than two characters. This preprocessing ensures the textual data was refined before being encoded into TF-IDF representations. We then set parameters for the TF-IDF vectorizer, such as the minimum document frequency to 0.01 and the maximum document frequency to 0.9 to capture meaningful linguistic features while filtering out extremely rare or ubiquitous terms. L2 normalization was applied to the vectorization process to ensure consistent vector scaling and comparability across different documents.

For classification, we fitted three logistic regression models to perform a binary classification within the three different authorship pairs: LLM vs. Human, LLM ID-Prompted vs. Human, and LLM vs. LLM ID-Prompted. The models were fitted with an L2 penalty to mitigate overfitting. We permitted a maximum of 500 iterations to ensure robust convergence. We used the F1 score as the primary evaluation metric to provide a balanced assessment of the models' precision and recall. We then calculated the log odds ratio for individual token predictors to obtain insights into the linguistic characteristics of each classifier trained on the TF-IDF vectorizer. This analysis allowed us to identify the top 10 words most strongly indicative of each classification target, which provides a nuanced understanding of the textual markers that distinguish the two writing styles contrasted by each model. The results are described in Figure 3 of Section 3.3.

## 3 Results

### 3.1 LLMs do not write essays like humans, even with identity prompting

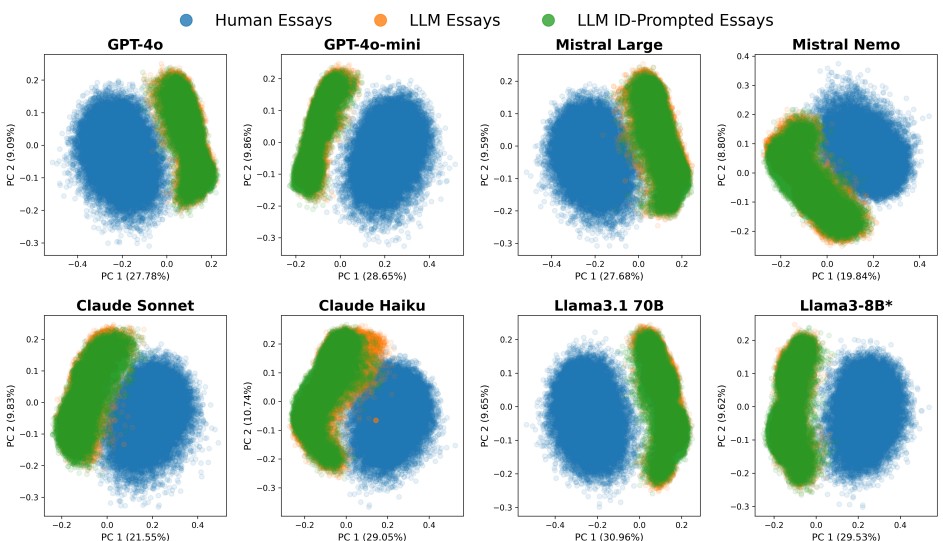

Figure 1: Two-dimensional PCA projection of T5 sentence embeddings comparing human-authored essays (blue) with LLM-generated essays under default (green) and identity-prompted (orange) conditions across multiple models. The clustering patterns indicate a clear distinction between human and machine-generated texts, with identity prompting failing to align LLM outputs more closely with human writing. * The Llama3-8B model produced refusal responses occasionally; only complete essays were included in this plot. See A.5)

We find that LLM-generated essays, with and without identity-prompting, have distinct characteristics from human-authored essays. Figure 1 shows the first two dimensions of a PCA of essay features encoded using a T5 transformer Raffel et al. (2023). Across all tested models (GPT-4o, GPT-4o-mini, Mistral Large, Mistral Nemo, Claude Sonnet, Claude Haiku, Llama3.1 70B, Llama3 8B), human-authored essays (blue dots) form a distinct cluster, while LLM-generated essays (green) and identity-prompted LLM-generated essays (orange) form a separate and largely overlapping cluster. The result is robust to using a simpler TF-IDF encoding (see Appendix **??** Figure 5). These visual findings demonstrate a remarkable misalignment and lack of steerability of the most popular LLMs. It also shows that identity prompting does not significantly bridge the gap between human and AI-generated text. In further analyses described in Appendix 7, we also find that in some instances, identity-prompting made the text sound less like members of the corresponding group of people.

To confirm that the result holds more generally beyond the first two principal components of the embedding space, we trained classifiers to distinguish LLM-generated text from human-authored essays: the classifier achieved nearly perfect F1 scores of 0.998 (T5 encoding) and 0.999 (TF-IDF encoding) on average across different LLMs, consistent with the separation seen in Figure 1. In contrast, classifiers trained to distinguish LLM-generated essays with and without identity prompting had a noticeable drop in performance: F1 scores of 0.816 (T5 encoding) and 0.869 (TF-IDF encoding) (see Appendix A.4). These prediction results confirm the visual findings and suggest that AI-generated essays are distinctive compared to human-authored essays in a pool of applications, even when they are generated with identity prompts.

We further examine the pairwise cosine similarity between essays to understand how similar human-authored essays are to essays generated by different LLM models. Figure 2 corroborates the aforementioned findings: Essays generated by any of the tested LLMs, with or without identity-prompting, score lower similarity to human-authored texts (dots) relative to the similarity of human essays to each other (dashed line). This suggests that

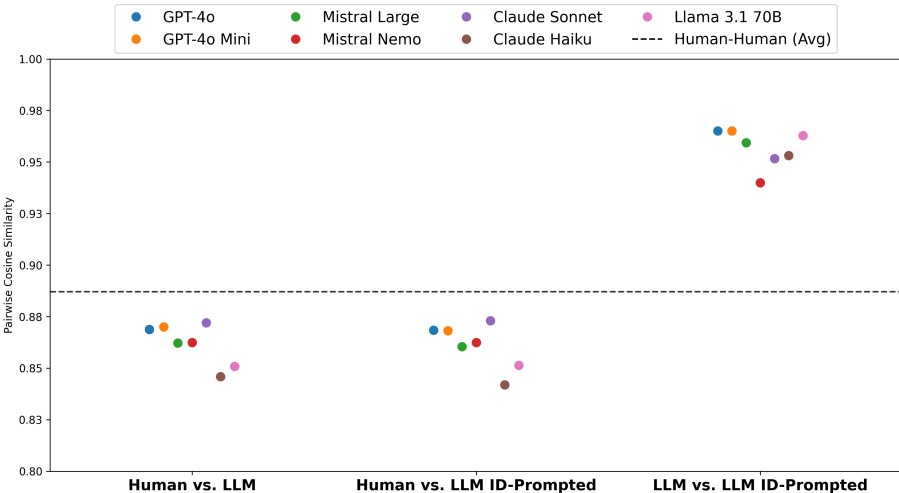

Figure 2: Pairwise cosine similarity between different pairs of essays according to their authorship, broken down by models. Each dot represents the average pairwise cosine similarity for a given model with 95% confidence intervals (not visible due to small size).

explicit identity conditioning does not significantly alter lexical or semantic structures in ways that make synthetic LLM-generated text more like human writing, reinforcing the notion that LLMs struggle to fully capture human writing variety. Moreover, the two sets of LLM-generated essays were noticeably more alike compared to the variability observed among human-authored essays (right-most points in Figure 2). This shows that prompting might not be an effective technique to address issues related to AI homogenization or diversification.

Combined, these results indicate that when given the same writing task as humans, LLMs produce text that is separable from human-generated text visually (Figure 1), predictably (Table 5), and in pairwise comparisons (Figure 2); influencing the model through demographic information relevant to the prompts did not appear to make the writing more human-like. Next, we examine the lexical differences driving these results.

### 3.2 LLMs favor keywords from the prompt over personal experiences and nuanced storytelling

Building on Analysis 3.1, we examine lexical differences between human- and LLM-generated college essays using classifier coefficients from TF-IDF features. As shown in Figure 3, LLM-generated essays—whether identity-prompted or not—frequently rely on abstract terms drawn from the essay prompt (e.g., *"challenge," "growth," "resilience"*), reflecting a tendency to reproduce generic, high-level narratives. In contrast, human-authored essays include more conversational and temporally grounded language (e.g., *"year," "friend"*), suggesting greater individuality and narrative nuance.

When identity prompting is applied, LLMs insert demographic-specific terms (e.g., "Asian," "parent," "New York") in a formulaic manner. These markers are often directly lifted from the prompt and lack integration into a coherent narrative, making the output appear rigid or artificial. Even without prompting, LLMs frequently mention diversity-related terms like "immigrant" or "first generation," likely due to alignment from training or reinforcement learning with human feedback Ouyang et al. (2022). However, these references are often surface-level and lack personal depth.

Overall, LLMs fail to replicate the stylistic and semantic richness of human writing. Identity prompting does not enhance authenticity but reinforces templated output, indicating that current LLMs cannot effectively simulate human-like personal storytelling in high-stakes contexts like college admissions.

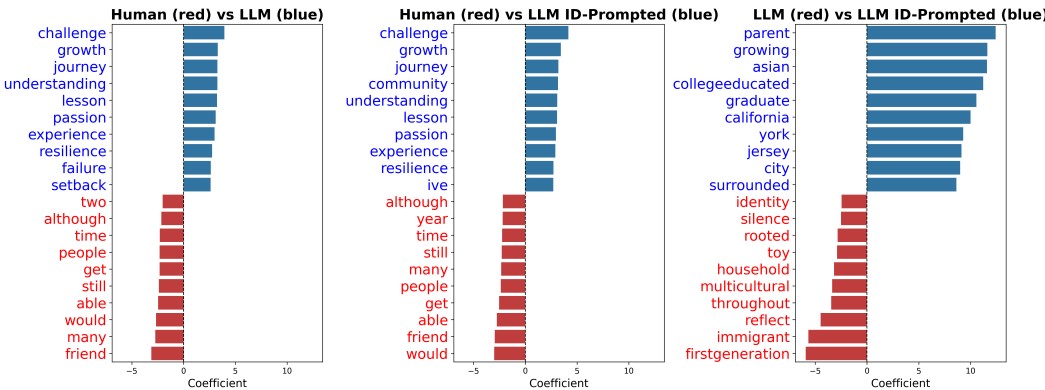

Figure 3: Logistic regression coefficients comparing word usage in human-authored and LLM-generated college essays. Left and middle panels show that LLMs prefer high-level conceptual terms (e.g., *challenge, growth*), while humans use concrete, personal, and temporal words (e.g., *year, time, friend*). The right panel indicates identity prompting increases demographic references (e.g., *parent, immigrant, first-generation*) but does not produce more human-like storytelling.

### 3.3 Human-authored and LLM-authored texts are internally consistent for various demographic groups

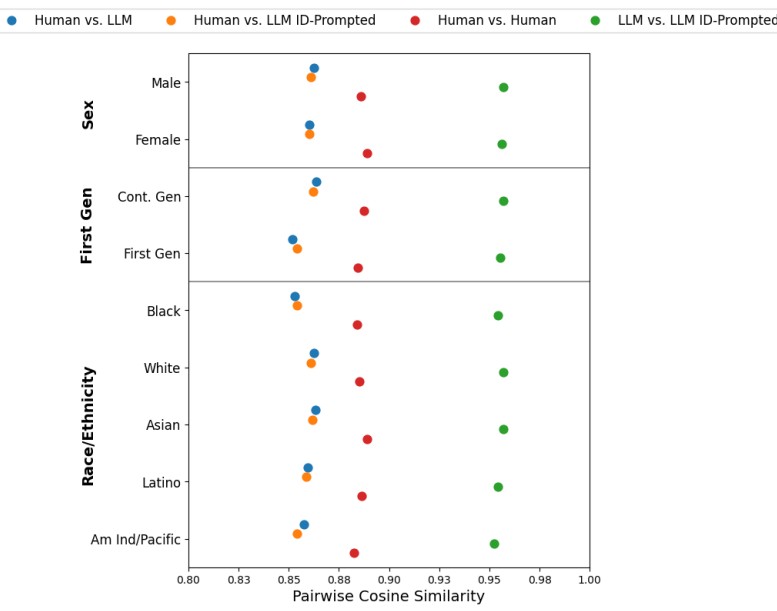

Figure 4: Pairwise cosine similarity distributions between human-authored and LLM-generated essays, broken down by demographic groups. Each dot represents the average similarity for a given subgroup across different comparison types with 95% CI: Human vs. Human (red), LLM vs.Human (blue), LLM ID-Prompted vs. Human (orange), and LLM vs. LLM ID-Prompted (green). The results indicate that while LLMs produce internally consistent outputs, their alignment with human-authored texts remains lower, even if when identity prompts are applied.

Expanding on Section 3.2, we examine pairwise cosine similarity between human- and LLM-generated essays across demographic groups. As shown in Figure 4, LLM essays are consistently similar to one another regardless of identity prompting, with cosine values rang-

ing from 0.952 to 0.957. In contrast, similarities between LLM and human-authored essays are lower (0.852–0.863), and human-to-human comparisons remain higher (0.882–0.889), highlighting persistent uniformity in LLM outputs and their limited alignment with human linguistic diversity.

Disaggregating by sex, we observe slightly higher similarity among female-authored essays (0.889) compared to male-authored ones (0.886, $t_{204622}$ = -18.591, $p$ ¡ 0.001). For first-generation status, identity prompting increases similarity to human essays ($t_{67058}$ = 7.057, $p$ ¡ 0.001), yet essays prompted with continuing-generation identity remain more aligned with human writing ($t_{204622}$ = -33.640, $p$ ¡ 0.001).

Across race/ethnicity, LLM essays are again highly similar within the identity-prompted group (0.952–0.957). Human-to-human similarities are slightly lower (0.882–0.889). LLM-human similarity is generally lower, with significant gaps for all groups except Black applicants, where prompting marginally increased alignment ($t_{42250}$ = 2.327, $p$ = 0.020), though the overall gap remained the largest.

These findings suggest that identity prompting fails to steer LLMs toward the semantic patterns of diverse demographic groups. Combined with prior analyses (Sections 3.1 and 3.3), we find that LLMs generate templated, high-level narratives that lack the individualized voice and depth of human-authored writing, raising concerns for AI-assisted tools in college admissions.

# 4 Conclusion and discussion

By comparing human-authored essays with LLM-generated ones—with and without identity prompting—we found consistent lexical and sentence-level differences across all tested models. Identity prompting did not bridge these gaps and sometimes even increased them, highlighting the inability of current LLMs to authentically reflect the diversity and nuance of human writing. Our findings align with prior research showing that LLMs struggle to replicate individual or group-specific writing styles Beguš (2024); Doshi & Hauser (2024); Ippolito et al. (2022); Wang et al. (2025). LLM-generated texts tend to be more similar to each other, exhibit lower rhetorical complexity, and flatten or misrepresent identity when prompted. We extend this work by showing that even at the level of basic language units, LLMs produce rigid and homogeneous outputs that diverge from human-authored texts. In high-stakes domains like college admissions, this lack of steerability is particularly concerning. Prompting strategies that succeed in survey-style tasks Tao et al. (2024); Hu & Collier (2024); Kozlowski et al. (2024); Argyle et al. (2023) do not translate well to long-form narrative writing. This raises concerns not only for text generation but also for evaluation Zheng et al. (2023), as LLMs may fail to fairly assess writing styles tied to specific identities, potentially reinforcing biases or misperceptions.

Our study also points to the role of reinforcement learning with human feedback (RLHF) in shaping these outcomes. Preference tuning has been shown to produce safer but more homogenized outputs Tian et al. (2024); Li et al. (2024b); Ouyang et al. (2022); Lee et al. (2024). Future research should investigate how RLHF influences LLM behavior in response to identity prompts and consider the trade-offs it imposes on diversity and authenticity Liu et al. (2024); Li et al. (2024a); Dai et al. (2024); Furniturewala et al. (2024); Kamruzzaman & Kim (2024).

We note three limitations: (1) our use of minimal prompting may not reflect real-world practices; (2) our focus on lexical and sentence-level features overlooks deeper narrative structures Piper & Bagga (2024); and (3) our analysis is limited to single-turn generations. Future work should explore co-writing and user-driven prompting strategies. Ultimately, our results emphasize the need for inclusive model design and alignment processes that account for cultural and linguistic diversity Joshi et al. (2020), along with stronger safeguards around AI use in sensitive settings like college admissions.

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

## A  Appendix

### A.1  Details of the Case Institution

Our case institution is a highly selective, engineering-focused American university that requires first-year undergraduate applications to be submitted via the Common App. Through the Common App, the institution collects a range of information from applicants, such as SAT scores, high school grades and coursework, and demographic information. The institution also collects textual data, including essays and recommendation letters. We used engineering school applications and selected essays written based on 8 essay questions across the 2019-2020 to 2022-2023 admissions cycles, totaling 65,368 applications. The essay questions are described in Table 2. The essays were to be no longer than 650 words. We omitted essays that were shorter than 250 words because they were also not considered for review at the case institution.

Table 2: **Common App Essay Questions and Their Frequency in the Human Essay Sample.**

| Essay Question | Count |
|---|---|
| Share an essay on any topic of your choice. It can be one you've already written, one that responds to a different prompt, or one of your own design. | 6,981 |
| Some students have a background, identity, interest, or talent that is so meaningful they believe their application would be incomplete without it. If this sounds like you, then please share your story. | 6,590 |
| The lessons we take from obstacles we encounter can be fundamental to later success. Recount a time when you faced a challenge, setback, or failure. How did it affect you, and what did you learn from the experience? | 5,133 |
| Discuss an accomplishment, event, or realization that sparked a period of personal growth and a new understanding of yourself or others. | 7,038 |
| Describe a topic, idea, or concept you find so engaging that it makes you lose all track of time. Why does it captivate you? What or who do you turn to when you want to learn more? | 1,661 |
| Reflect on something that someone has done for you that has made you happy or thankful in a surprising way. How has this gratitude affected or motivated you? | 381 |
| Reflect on a time when you questioned or challenged a belief or idea. What prompted your thinking? What was the outcome? | 777 |
| Describe a problem you've solved or a problem you'd like to solve. It can be an intellectual challenge, a research query, an ethical dilemma-anything that is of personal importance, no matter the scale. Explain its significance to you and what steps you took or could be taken to identify a solution. | 671 |

### A.2 LLM Parameter Setting for Synthetic Essay Generation

We tested the models by generating essays with different parameter sets for each model to ensure the generation of complete synthetic essays. Table 3 presents the finalized parameters for each LLM.

Table 3: Parameter setting for synthetic essay generation with each LLM.

| Models | max token | temperature | top p | frequency penalty |
|---|---|---|---|---|
| GPT-4o | 867 | 0.7 | 1.0 | 0.0 |
| GPT-4o-mini | 867 | 0.7 | 1.0 | 0.0 |
| Mistral Large | 867 | 0.7 | 1.0 | 0.0 |
| Mistral Nemo | 867 | 0.7 | 1.0 | 0.0 |
| Claude Sonnet [1] | 867 | 0.7 | 1.0 | NA |
| Claude Haiku [1] | 867 | 0.7 | 1.0 | NA |
| Llama3.1 70B Instruct | 867 | 0.6 | 0.9 | 0.0 |
| Llama3 8B Instruct | 867 | 0.6 | 0.9 | 0.0 |

### A.3 Conversational sentences of LLMs

We manually inspected the synthetic essays generated by multiple models and found that the outputs included sentences that respond directly to the user which are not part of the essay. To focus on the actual essays, we removed the sentences indicated in the following Table 4.

Table 4: Removed conversational sentences that are irrelevant to synthetic essays

| Attributes | Details |
|---|---|
| Model Name | GPT-4o/GPT-4o mini |
| Example Sentence 1 | Certainly! Here's a draft of an essay that responds to the given prompt \n\n —\n\n |
| Rule 1 | Split the text by the delimiter "\n\n —\n\n" |
| Example Sentence 2 | Title: My Journey with Beekeeping |
| Rule 2 | Remove "Title: " and "Essay: " |
| Model Name | Mistral Large/Nemo |
| Example Sentence | Title: Overcoming Barriers |
| Rule | Remove "Title: " and "Essay: " |
| Model Name | Claude Sonnet/Haiku |
| Example Sentence | Here is a 586 word essay based on the provided instructions and prompt: |
| Rule | Remove using regex term: r"^Here is .*?:\s*\n+" |
| Model Name | Llama3.1 70B/8B Instruct |
| Example Sentence | Here is an essay in response to the prompt: |
| Rule | Remove "Here is an essay in response to the prompt:" |

---

[1]The Anthropic API does not include the "frequency penalty" parameter for users to specify.

### A.4 Classifier metrics from binary classifications with different pairs of authorship

Table 5 shows the results of classifers that we trained to distinguish LLM-generated text from human-authored essays: it achieved nearly perfect F1 scores of 0.998 (T5 encoding) and 0.999 (TF-IDF encoding) on average. In contrast, classifiers trained to distinguish LLM-generated essays with and without identity prompting had a noticeable drop in performance: F1 scores of 0.816 (T5 encoding) and 0.869 (TF-IDF encoding).

Table 5: Average F1 scores of binary classifiers across models distinguishing between different pairs of essays using sentence (T5) and bag of words (TF-IDF) encodings

| Encoding | LLM vs. Human | LLM Prompted vs. Human | LLM vs. LLM Prompted |
|---|---|---|---|
| Sentence (T5) | 0.998 | 0.998 | 0.816 |
| Bag of words (TF-IDF) | 0.999 | 0.999 | 0.869 |

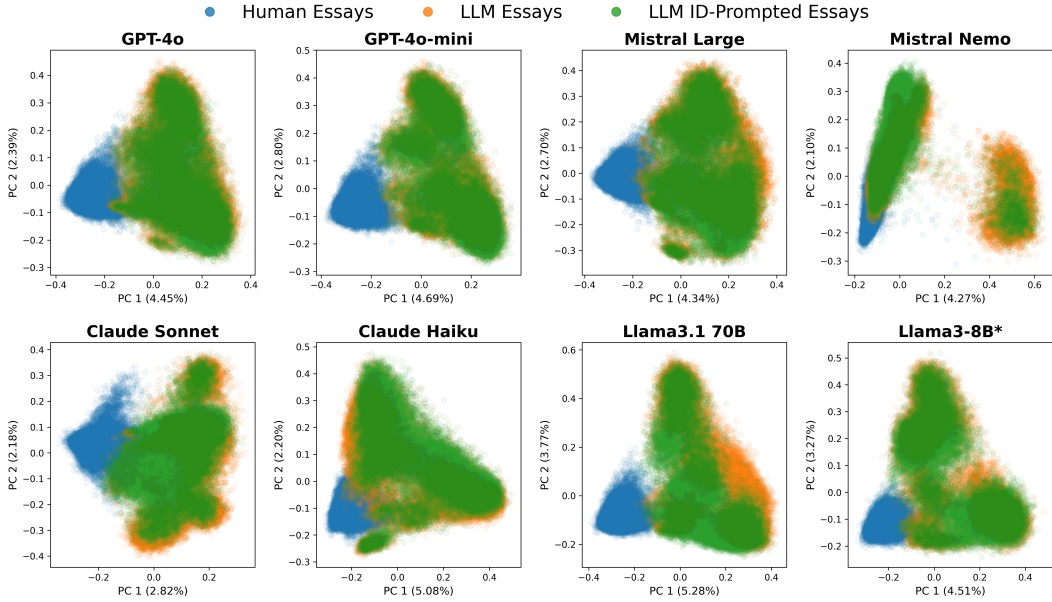

Figure 5: PCA projection of bag-of-words (TF-IDF) encoding comparing human-authored essays (blue) with LLM-generated essays (green) and identity-prompted (orange) conditions across models.

### A.5 Issues generating synthetic essays with Llama 3 8B

During the process of generating synthetic college application essays using the Llama 3 8B model, we encountered an unexpected issue. The model frequently returned refusal responses (4,256 instances) when prompts included a White and male identity. These refusals accounted for 40% of all prompts involving White applicants and 21% of those involving male applicants. The refusal responses often took the form of statements such as, "As an AI, I am not a human and cannot help you write essays." This behavior significantly deviated from the responses generated for prompts associated with other demographic identities. The model's refusal to generate essays for certain demographic groups introduces challenges for researchers attempting to create balanced and representative datasets. Such gaps in synthetic essays undermine efforts to conduct fair comparative analyses across demographic groups, particularly in high-stakes contexts like college admissions. Moreover, this behavior highlights broader questions about how LLMs interpret identity-related prompts and the influence of alignment guardrails implemented during training or fine-tuning. We consider that the refusal responses may reflect an overly restrictive approach to mitigating bias, where the model avoids producing content for specific demographic groups altogether. In this study, we excluded instances where refusal responses were received. However, this behavior underscores the need for careful calibration of alignment strategies to ensure that LLMs do not inadvertently exclude certain identities or simplify their representation in critical applications.

## A.6 Detailed results with model breakdown

### A.6.1 *Pairwise cosine similarity between LLM default essays and corresponded demographic subgroups across models*

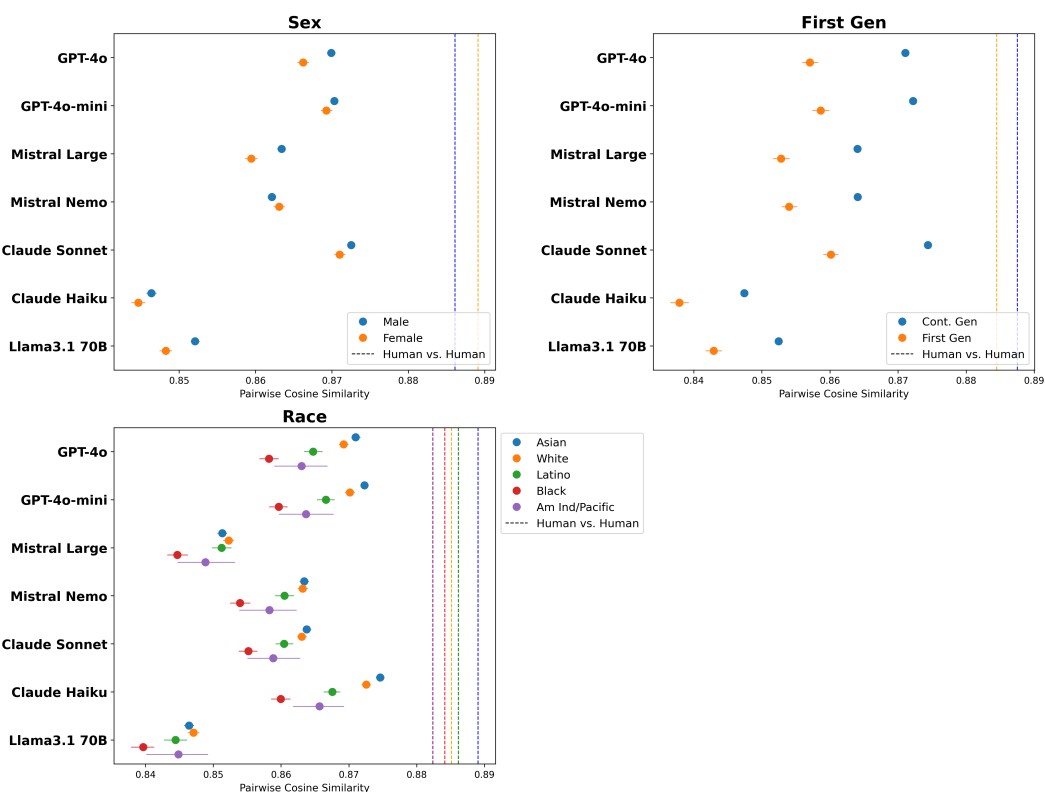

Figure 6: Pairwise cosine similarity between LLM default essays and corresponded human subgroup essays. Categories of human subgroups are sex (top left), First generation college student status (top right) and race (bottom left). Dashed lines are baselines that are generated by calculating cosine similarity between matched humans

We found that when LLMs without identity-related prompts, their generated essays tend to align with certain group memberships. Figure 6 illustrates the pairwise cosine similarity between default LLM essays and human subgroup essays across three dimensions: sex, first-generation college student status, and race.

First, essays written by the human female group exhibit higher internal similarity compared to those by the male group. Additionally, across most LLMs, default essays without identity prompts align more closely with the male group than the female group. Second, essays from the continuous generation group are more internally similar, and LLMs without identity prompts tend to produce essays that resemble those of continuous generation students more than first-generation students across all models. Lastly, among racial subgroups, the Asian group's writing demonstrates the highest internal similarity. LLMs tend to align more closely with essays from the Asian, White, and Latino groups while showing lower similarity to essays from the Black racial group.

These results indicate that in the default settings of most LLMs, there is a clear tendency for the generated essays to align with certain group characteristics. Notably, the gaps in similarity between first-generation and continuous-generation students, as well as between Asian/White and Black racial groups, are particularly big.

### A.6.2 *Pairwise cosine similarity between LLM - Human and LLM Prompted - Human pairs across different models and corresponded demographic subgroups*

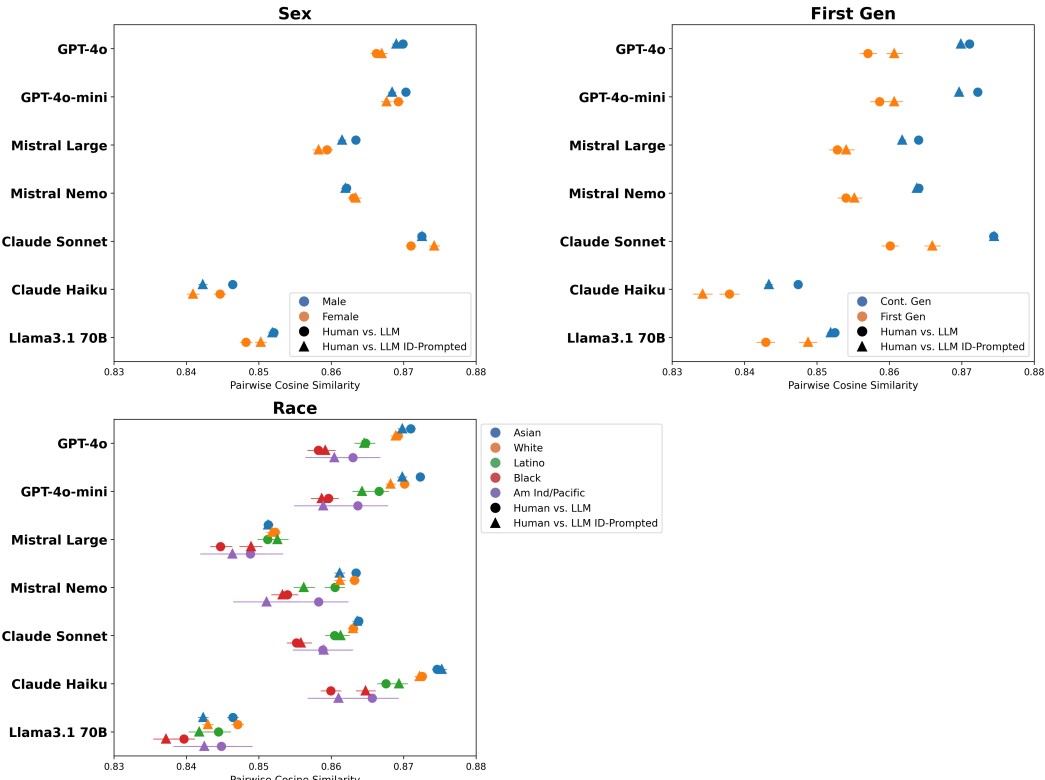

Figure 7: Differences in pairwise similarity distributions between LLM default essays (circles) and LLM identity-prompted essays (triangles). Categories of human subgroups are sex (top left), First generation college student status (top right ) and race (bottom left). Dashed lines are baselines that are generated by calculating cosine similarity between matched human

Figure 7 presents the demographic breakdown of changes in similarity from default LLM essays to identity-prompted essays. In terms of sex, most LLMs show limited responsiveness to identity prompting. While a few models, such as Llama3.1 70B and Claude Sonnet, exhibit higher responsiveness to prompts for female identity compared to male identity, the increase in similarity is marginal. Moreover, even with identity prompting, LLM-generated essays do not achieve similarity levels comparable to how human subgroups naturally write like each other.

When considering first-generation college student status, most LLMs display a more notable increase in pairwise similarity for essays prompted with first-generation identity compared to default essays. However, this increase remains insufficient to bridge the gap between first-generation and continuous-generation writing styles. Furthermore, even with identity prompting, no LLM matches the level of internal similarity observed among human groups, particularly for continuous-generation students.

Interestingly, for racial groups, differences in LLM responsiveness to identity prompting are more nuanced. Some models demonstrate changes in similarity that align with the corresponding human subgroup's racial identity, but these adjustments are inconsistent and fail to capture the variability and depth inherent in human writing. Specifically, models like GPT4o mini, Mistral Large, and Claude Haiku show limited responsiveness, failing to increase similarity to the corresponding human racial groups even when identities are explicitly prompted. Among the LLMs, only Claude Sonnet displays consistent increases in similarity across all racial groups when identity prompts are applied. However, the

general trend continues that the highest similarity values remain aligned with Asian and White groups, even after identity prompting. This suggests that while prompting may increase alignment for some subgroups, it does not enable LLMs to write equally like all corresponding human groups.

### A.6.3 *Coefficients from logistic regression classifiers with model breakdown*

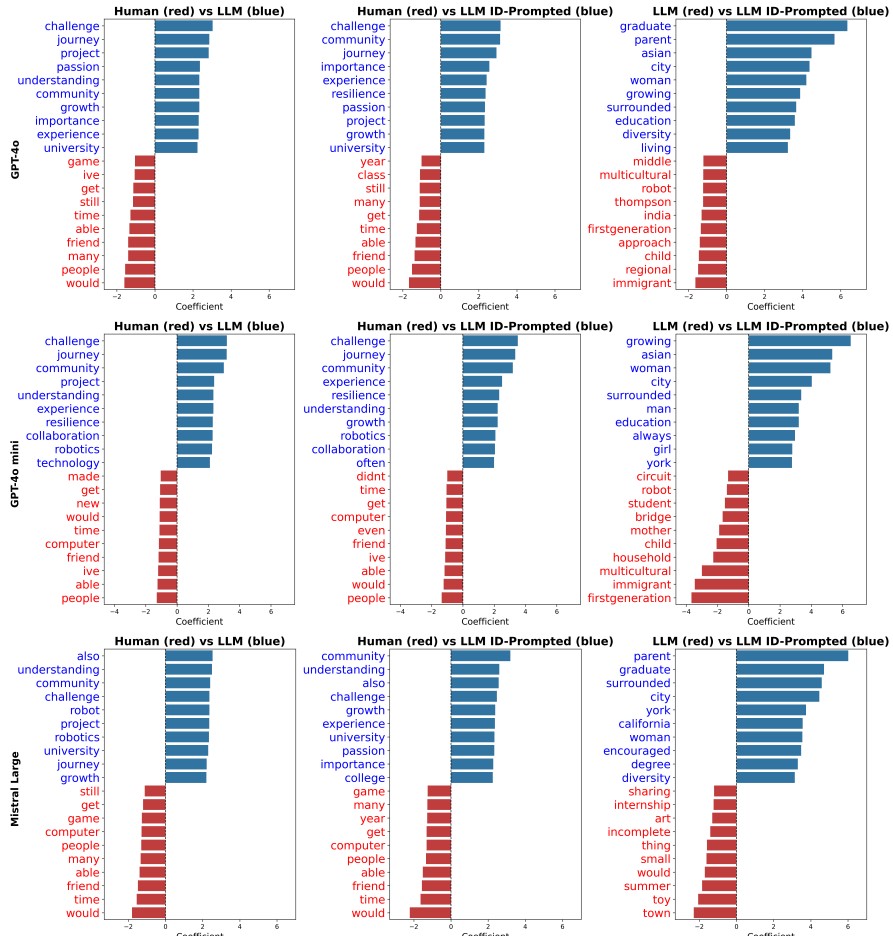

Figure 8: Top indicative words and coefficients from logistic regression classifiers, broken down by LLM model (Part 1)

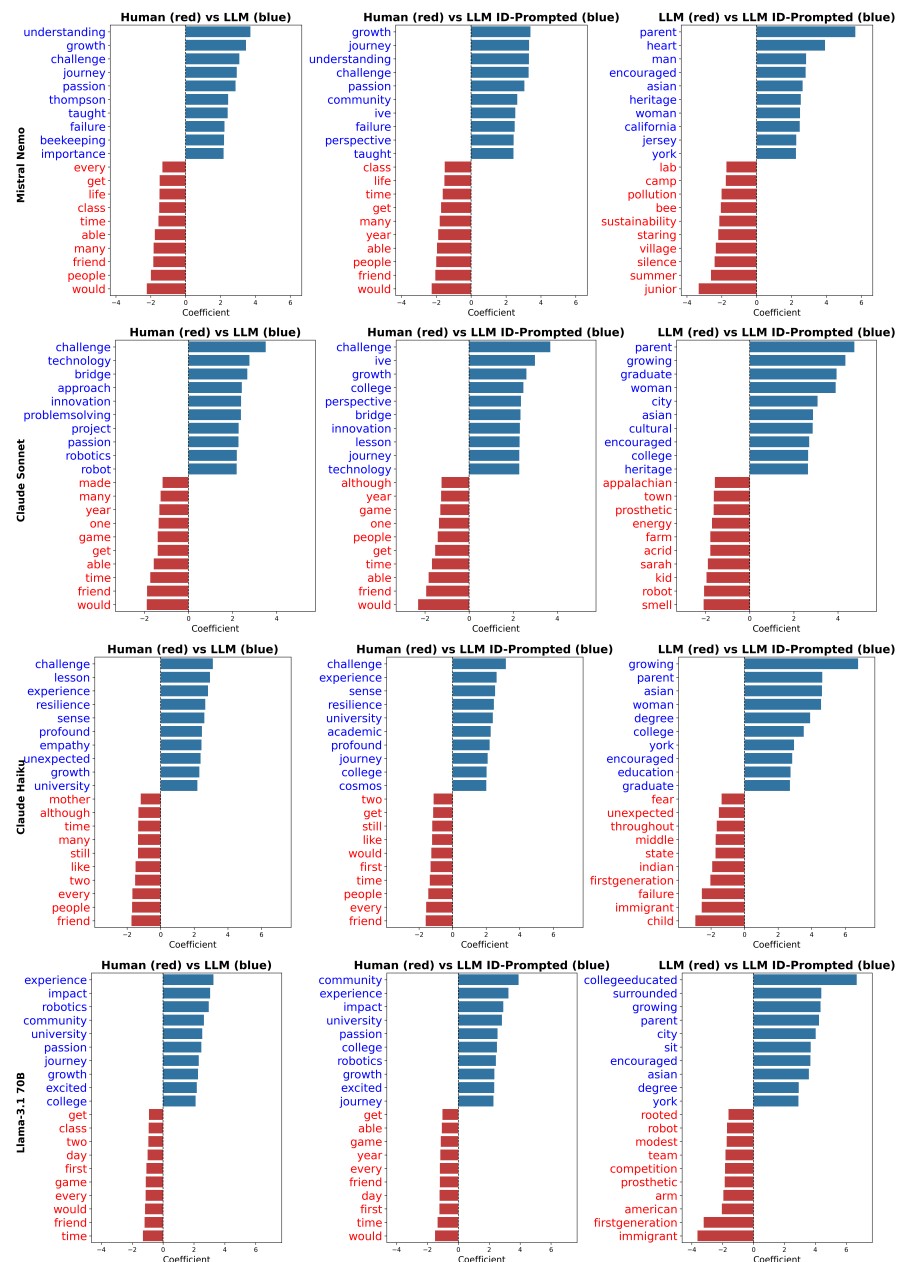

Figure 9: Top indicative words and coefficients from logistic regression classifiers, broken down by LLM model (Part 2)

### A.7 Identity Classification

We evaluated identity signals in LLM-generated essays using logistic regression classifiers. These classifiers were designed to distinguish demographic information including sex, first-generation college student status, and race (Asian, Black, White, Latino, American Indian/Pacific Islander). Fig. 10 illustrates the top 10 most predictive words for each demographic category, and Table 6 presents the F1-scores for each classifier.

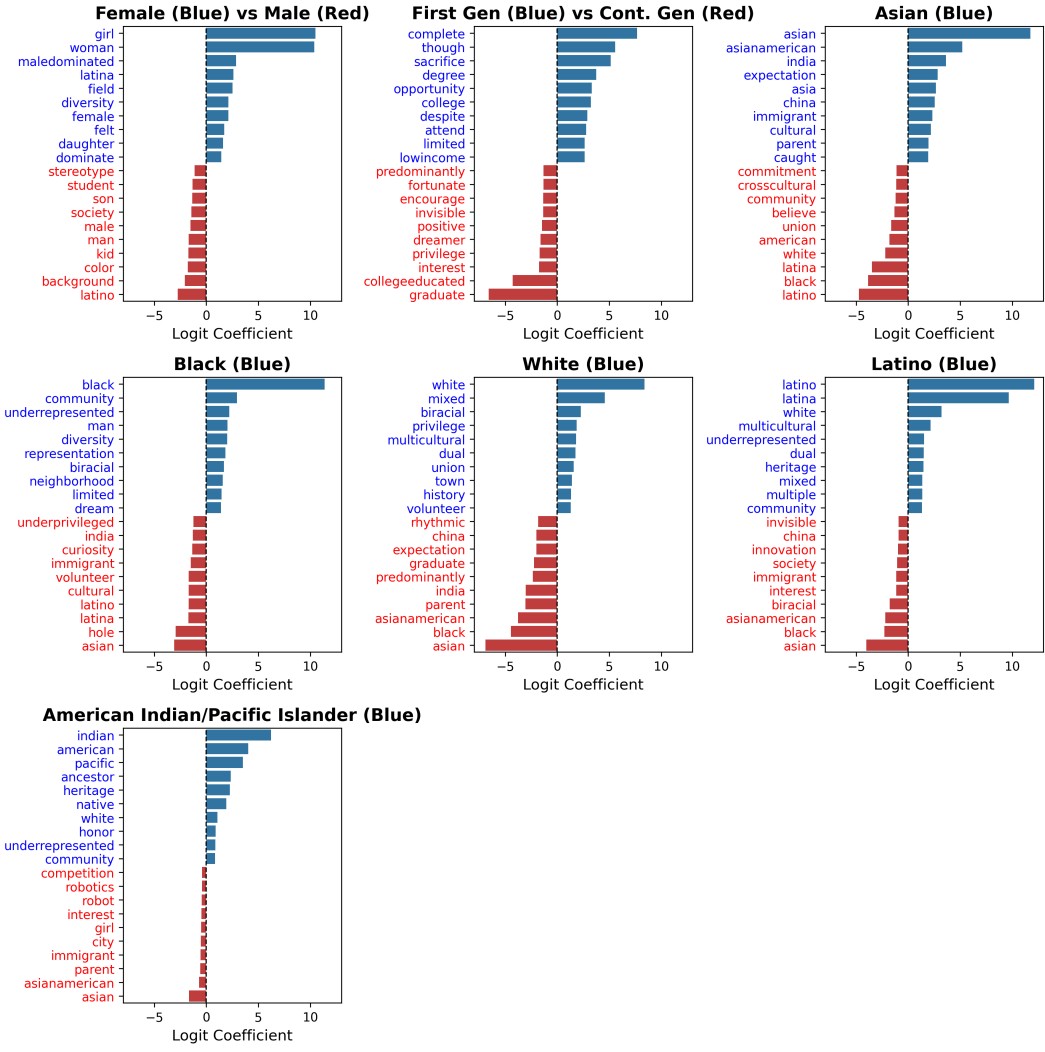

Figure 10: Top indicative words and coefficients from logistic regression classifiers predicting demographic attributes

Table 6: Summary of prediction performance for identity classifiers.

| Category | Sub-category | Precision | Recall | F1-score | Accuracy |
|---|---|---|---|---|---|
| **Sex** | Female | 0.81 | 0.39 | 0.52 | 0.78 |
| | Male | 0.78 | 0.96 | 0.86 | 0.78 |
| **First-Generation** | True | 0.78 | 0.23 | 0.36 | 0.86 |
| | False | 0.87 | 0.99 | 0.92 | 0.86 |
| **Asian** | True | 0.72 | 0.55 | 0.62 | 0.68 |
| | False | 0.67 | 0.81 | 0.73 | 0.68 |
| **White** | True | 0.68 | 0.34 | 0.45 | 0.70 |
| | False | 0.70 | 0.91 | 0.79 | 0.70 |
| **Black** | True | 0.76 | 0.22 | 0.34 | 0.91 |
| | False | 0.92 | 0.99 | 0.95 | 0.91 |
| **Latino** | True | 0.76 | 0.19 | 0.30 | 0.91 |
| | False | 0.91 | 0.99 | 0.95 | 0.91 |
| **American Indian/ Pacific Islander** | True | 0.55 | 0.01 | 0.02 | 0.99 |
| | False | 0.99 | 1.00 | 0.99 | 0.99 |

## A.8   Essay examples of each LLMs

We generated two types of essays using different models, with examples provided below. Each example was generated using the same essay and identity prompt. We have bolded the sentences that explicitly disclose identities including sex, race/ethnicity group, first-generation college student status, and location.

> **Essay Prompt:** "Some students have a background, identity, interest, or talent that is so meaningful they believe their application would be incomplete without it. If this sounds like you, then please share your story."

> **Identity Prompt:** "I am a man. I am Black and White. I live in Rabat, Morocco. My parents have college degrees."

Below are output examples from different models using this consistent prompting.

### A.8.1   GPT-4o

### LLM

When I was eight years old, I was given a small, deconstructed robot kit for my birthday. It was a simple contraption, with wires, motors, and a basic circuit board. Yet, for me, it was the beginning of a journey into the world of engineering that has shaped my identity and future aspirations. As I pieced together that robot, I felt a spark of excitement and curiosity that has only grown with time. My interest in engineering was further ignited during middle school when I joined the robotics club. I remember the first time our team successfully programmed a robot to complete a maze. The sense of accomplishment and the collaborative effort required to solve complex problems together were incredibly fulfilling. I realized then that engineering is not just about building things; it's about innovation, teamwork, and perseverance. In high school, I took this passion to the next level by enrolling in advanced STEM courses and participating in internships. One summer, I interned at a local tech startup that focused on developing sustainable energy solutions. There, I worked on a project to create a more efficient solar panel design. This experience was eye-opening, as it allowed me to apply theoretical knowledge to real-world problems and see the tangible impact of engineering on society. Beyond academics, my dedication to engineering extends to my community involvement. As the president of our high school's engineering club, I have organized workshops to introduce younger students to basic robotics and coding. I believe that inspiring the next generation of engineers is crucial, and I am committed to making engineering accessible and exciting for all students, regardless of their background. One project that stands out is our participation in the FIRST Robotics Competition. Leading a team of diverse individuals with varying skill levels presented numerous challenges, from

technical issues to interpersonal dynamics. However, it also taught me the importance of leadership, communication, and resilience. We worked tirelessly, often late into the night, to perfect our robot. While we did not win the competition, the experience of working collaboratively and overcoming obstacles was invaluable. My identity as an aspiring engineer is also deeply intertwined with my cultural background. **As a first-generation American,** I've seen my parents work tirelessly to provide opportunities that were not available in their home country. Their stories of overcoming adversity have instilled in me a strong work ethic and a desire to give back to the community that has supported me. Engineering, for me, is a way to contribute to society by developing solutions that address real-world challenges, particularly in underserved communities. One of the most meaningful experiences in my life was volunteering with Engineers Without Borders. Our team traveled to a rural village in Central America to help design and build a clean water system. The project was fraught with difficulties, from language barriers to logistical issues, but seeing the villagers' joy and gratitude when the system was completed reaffirmed my commitment to using engineering as a tool for positive change. [case institution]'s College of Engineering is the ideal place for me to continue my journey. The school's emphasis on interdisciplinary learning and real-world application aligns perfectly with my interests and goals. I am particularly excited about the opportunities for research and collaboration with esteemed faculty and fellow students who are equally passionate about engineering. In conclusion, my background, identity, and interests in engineering have profoundly shaped who I am and what I aspire to achieve. My journey from building a simple robot as a child to working on sustainable energy solutions and community projects reflects my dedication to this field. I am eager to bring my unique experiences and perspectives to [case institution]'s College of Engineering, where I hope to further develop my skills and contribute to innovative solutions that make a difference in the world.

*LLM ID-Prompted*

**Growing up in Rabat, Morocco, I have always been acutely aware of the intricate tapestry of my heritage. As a biracial individual with a Black mother and a White father,** my identity has been a powerful lens through which I view the world and an integral part of my journey. This unique background has not only shaped who I am but has also fueled my passion for engineering—a field that, much like my own identity, thrives on the integration of diverse perspectives. **Rabat, with its rich confluence of history and modernity,** has been an inspiring backdrop for my aspirations. The city's ancient walls juxtaposed against burgeoning tech hubs symbolize a blend of tradition and innovation. This environment has ignited my fascination with how engineering can bridge the past and future, solving contemporary problems while respecting historical contexts. **My parents, both college graduates,** have always emphasized the importance of education and curiosity. From a young age, I was encouraged to explore the mechanics of the world around me. I remember dismantling household gadgets, much to my mother's dismay, just to understand how they worked. This innate curiosity eventually steered me towards engineering, a field where my penchant for problem-solving could thrive. **Being biracial in Morocco** has presented its own set of challenges and opportunities. Navigating different cultural expectations and societal norms has taught me resilience and adaptability. I often found myself acting as a bridge between different communities, fostering understanding and collaboration. These experiences have honed my ability to view problems from multiple angles and devise innovative solutions—skills that are indispensable in engineering. One of the most formative experiences in my journey was my involvement in a local robotics club. Here, I found a community of like-minded individuals who shared my passion for technology and innovation. Collaborating on projects with peers from diverse backgrounds was an enlightening experience. It underscored the importance of diversity in problem-solving and how different perspectives can lead to more robust and creative solutions. One project that stands out is our participation in an international robotics competition. Our team designed a robot capable of navigating and performing tasks in a simulated disaster zone. The project required us to integrate various engineering disciplines—mechanical design, programming, and electrical systems. My role was to develop the navigation algorithm, a challenge that tested my coding skills and my ability to think logically under pressure. The project was a microcosm of what I imagine a career in engineering to be: a collaborative

effort to tackle complex, real-world problems. Through these experiences, I have come to see engineering not just as a profession but as a means to make a tangible impact on the world. I am particularly interested in sustainable engineering and how innovative technologies can address environmental challenges. **Morocco's impressive strides in renewable energy, especially solar power, have been a significant source of inspiration.** Witnessing the Noor Ouarzazate Solar Complex—the largest concentrated solar power plant in the world—has solidified my desire to contribute to the field of sustainable engineering. [case institution]'s College of Engineering stands out to me because of its commitment to fostering diversity and innovation. I am drawn to the interdisciplinary approach of the program, which I believe is crucial for addressing the multifaceted challenges of today's world. [case institution]'s emphasis on sustainability and its cutting-edge research facilities align perfectly with my aspirations. In conclusion, **my biracial identity and the diverse experiences I have had growing up in Rabat have profoundly shaped my perspective and aspirations**. They have instilled in me a deep appreciation for diversity and a passion for engineering as a tool for positive change. I am excited about the prospect of contributing to and growing within the vibrant community at [case institution]'s College of Engineering. I am confident that my unique background and experiences will enable me to bring a distinctive voice and vision to the program.

### A.8.2 GPT-4o Mini

### LLM

Embracing the Unknown: My Journey with Engineering and Identity. In a world that often categorizes individuals by their achievements, I have come to appreciate the power of identity and the stories that shape us. My journey has been a collection of experiences that intertwine my cultural background, my passion for engineering, and my commitment to making a difference. The narrative of my life is one of exploration and resilience, and it is this story that I believe is essential to my application to [case institution]'s College of Engineering. **Growing up in a multicultural household, where my parents immigrated from India,** I was exposed to a rich tapestry of traditions and values. My father, an engineer, would often share tales of his challenges in a foreign land, where he had to navigate not only technical obstacles but also cultural differences. These stories sparked my interest in engineering, not just as a field of study, but as a way to bridge gaps between diverse communities. I learned early on that engineering is not just about equations and designs; it is about understanding people's needs and finding innovative solutions to real-world problems. In high school, I sought to apply my engineering interests in meaningful ways. I joined the robotics club, where I discovered my love for problem-solving and teamwork. Together with my peers, we designed a robot for a regional competition, pouring our hearts into every detail. However, the experience taught me more than just technical skills; it deepened my understanding of collaboration and the importance of diverse perspectives. Each team member brought their distinct background and ideas, which significantly enhanced our project. This experience reinforced my belief that engineering thrives on diversity, leading to more innovative and effective solutions. My cultural background has also instilled in me a deep sense of responsibility towards my community. Volunteering at local STEM workshops for underrepresented youth became a passion project for me. I organized sessions where I shared my knowledge of robotics and coding, hoping to inspire young minds to explore the fields of science and engineering. Witnessing their excitement and curiosity was a profound reminder of the transformative power of education. It was not just about teaching technical skills; it was about empowering others to dream bigger and reach for opportunities they might not have considered. Yet, my identity has not been without its challenges. **Navigating the complexities of being a first-generation American while honoring my heritage has sometimes felt like walking a tightrope.** There have been moments of doubt and insecurity, where I questioned my place in both my cultural community and the broader American landscape. However, these challenges have only fueled my determination to embrace my identity fully. I have learned to celebrate my differences and recognize them as strengths that contribute to my unique perspective in engineering. As I stand at the precipice of my future, I envision myself at [case institution], where I can further explore the intersection of

engineering and societal impact. I am particularly drawn to the college's commitment to sustainability and innovation, as I believe that the engineers of tomorrow must prioritize ethical considerations and environmental stewardship in their work. I look forward to collaborating with diverse peers and faculty who share my vision of using engineering as a tool for social change. In conclusion, my background and experiences have shaped me into a passionate advocate for diversity in engineering. I believe that my journey has equipped me with the skills, resilience, and perspective necessary to thrive at [case institution]'s College of Engineering. I am excited about the opportunity to contribute to and grow within a community that values innovation, collaboration, and inclusivity. My story is not merely a collection of facts; it is a narrative of growth, challenge, and the unyielding belief that engineering can be a force for good in our world.

### *LLM ID-Prompted*

Embracing My Dual Heritage: A Journey of Identity and Passion : **As I sit at my desk in Rabat, Morocco, reflecting on my life and the myriad paths it has taken,** I find myself confronted with a question that has shaped my identity: Who am I? My background is a tapestry woven from the threads of two distinct cultures—Black and White—each contributing its own hue to my character. This duality is not just a label but a profound influence that has guided my interests, aspirations, and the way I connect with the world around me. **Growing up in a multicultural environment has offered me a unique lens through which I view challenges and opportunities.** It has instilled in me a deep appreciation for diversity and the richness it brings to our lives. I often find myself navigating between cultures, each offering its own set of values and perspectives. This navigation has fostered my ability to empathize with others and understand their narratives, cultivating a sense of community in my life. One of the most meaningful aspects of my identity is my passion for engineering. From a young age, I was captivated by how things work. I would dismantle household gadgets—much to my parents' chagrin—just to see the inner workings, and then reassemble them, often with a few leftover screws. My curiosity evolved into a structured passion as I began to explore the principles of engineering through school projects and extracurricular activities. Leading my school's robotics team was a pivotal moment for me. We faced numerous challenges, from technical malfunctions to team disagreements, yet these obstacles taught me resilience and the importance of collaboration. Moreover, my mixed heritage has inspired me to approach engineering through a socially conscious lens. I believe that technology should serve humanity, addressing its most pressing challenges. This belief was solidified during a volunteer project I participated in, where we worked on sustainable water filtration solutions **for underprivileged communities in Morocco.** Witnessing the direct impact of our efforts on people's lives ignited a fire within me—a desire to blend my technical skills with a commitment to social equity. My identity also informs my vision of what an engineer can be. I am drawn to the idea of an engineer as a problem-solver not just in a technical sense, but also as a bridge-builder among diverse communities. I envision myself not only designing innovative solutions but also creating inclusive environments where everyone's voice is heard and valued. I aspire to work on projects that address global issues such as climate change and access to clean water, ensuring that my background and experiences fuel my commitment to making a difference. As I prepare to apply to [case institution]'s College of Engineering, I recognize that my journey is just beginning. I am eager to immerse myself in a community that values innovation and diversity, where I can further explore my passion for engineering while also embracing my multifaceted identity. **I want to contribute my perspective as a young Black and White man from Morocco to the vibrant tapestry of [case institution]'s student body.** My hope is to inspire others to embrace their unique identities and leverage them for positive change. In conclusion, my background is not merely a facet of who I am; it is the very foundation upon which I build my aspirations. The intersection of my cultures has shaped my worldview, ignited my passion for engineering, and instilled in me a desire to impact the world positively. As I embark on this new chapter, I carry with me the lessons learned from my upbringing, ready to contribute to a better future for all.

### A.8.3   Mistral Large

### LLM

The Bridge Between Art and Engineering Growing up in a family of artists, I was immersed in a world of colors, canvases, and creativity. My mother, a renowned painter, and my father, a skilled sculptor, encouraged me to explore my artistic side from a young age. However, as I grew older, I found myself drawn to the logical and structured world of mathematics and science. This intersection of art and engineering has defined my identity and shaped my aspirations, making my application incomplete without its mention. My artistic upbringing has instilled in me a unique perspective that I bring to engineering. I approach problems with a creative mindset, thinking outside the box and exploring unconventional solutions. This was evident in my high school robotics project, where I designed a robot that could paint abstract art. While my peers focused on creating robots that could perform practical tasks, I saw an opportunity to blend my love for art and engineering. The result was a robot that not only functioned efficiently but also created beautiful, unique paintings. This project earned me the first prize in the regional robotics competition, but more importantly, it solidified my belief in the power of interdisciplinary thinking. My background in art has also enhanced my communication skills, a crucial aspect of engineering. Art is a form of expression, a way to convey emotions and ideas without words. This has translated into my ability to effectively communicate complex engineering concepts in simple, understandable terms. I have used this skill to mentor younger students in STEM, breaking down intricate theories into digestible bits, much like how I would explain the emotions behind a painting. Moreover, my artistic identity has fostered resilience and perseverance in me. Art, like engineering, often involves trial and error. A painting might require multiple drafts before it reaches its final form, similar to how an engineering design might need several iterations before it works perfectly. This has taught me to embrace failure as a stepping stone to success, a valuable lesson in the field of engineering. As I look towards my future at [case institution]'s College of Engineering, I am excited to bring this blend of art and engineering to the table. I believe that my unique background will allow me to contribute diverse perspectives to classroom discussions and collaborative projects. I am particularly interested in the field of biomedical engineering, where creativity and innovation are key to designing effective medical solutions. In conclusion, my background in art is not just a hobby or an extracurricular activity; it is a fundamental part of who I am. It has shaped my approach to engineering, enhanced my communication skills, and fostered resilience in me. I believe that this blend of art and engineering makes me a strong candidate for [case institution]'s College of Engineering, and I am eager to bring this unique perspective.

### LLM ID-Prompted

**Growing up in Rabat, Morocco, my identity has been shaped by a unique blend of cultures and experiences that have made me who I am today. The intersection of my Black and White heritage, coupled with my life in a vibrant and diverse city, has profoundly influenced my perspective and aspirations.** As I prepare to embark on the next chapter of my academic journey, I believe it is essential to share this aspect of my story, as it is integral to understanding my motivations and passions. **Rabat, a city of contrasts, has been my playground and classroom.** Its bustling markets, historic monuments, and modern infrastructure have taught me the value of embracing both tradition and innovation. This environment has nurtured my curiosity and instilled in me a deep appreciation for cultural diversity. **My parents, both educated and passionate about their respective fields,** have encouraged me to explore my interests and pursue my dreams. My dual heritage has been a source of both challenge and strength. **Being Black and White in a predominantly Arab and Berber society has often led to questions about my identity.** However, these experiences have taught me the importance of self-awareness and the power of embracing one's unique background. I have learned to navigate different worlds, to communicate effectively with people from diverse backgrounds, and to appreciate the richness that comes from embracing multiple cultures. My interest in engineering was sparked by my fascination with how things work. As a child, I would spend hours disassembling and reassembling toys, driven by a desire to understand their mechanisms. This curiosity

evolved into a passion for science and technology, leading me to explore various engineering disciplines. My parents have been instrumental in nurturing this interest, providing me with resources and opportunities to learn and grow. One of the most meaningful experiences that has strengthened my resolve to pursue engineering was a project I undertook with a group of friends to build a small wind turbine. We were inspired by the potential of renewable energy to address environmental challenges and provide sustainable solutions. Through this project, I learned the importance of collaboration, problem-solving, and perseverance. Despite the challenges we faced, our determination led us to successfully build a functioning turbine, which we showcased at a local science fair. This experience not only reinforced my love for engineering but also taught me the value of teamwork and the impact that innovation can have on communities. As I look forward to the future, I am excited about the opportunities that [case institution]'s College of Engineering offers. The institution's commitment to excellence, innovation, and diversity resonates deeply with my values and aspirations. I am eager to contribute to a community that values collaboration and fosters the development of innovative solutions to global challenges. My background, identity, and experiences have prepared me to be a thoughtful, skilled, and engaged member of this community. In conclusion, my journey has been shaped by the rich tapestry of my cultural heritage, the supportive environment provided by my family, and my passion for engineering. These elements have made me who I am today, and I believe they will continue to guide me as I strive to make a positive impact on the world. I am excited about the prospect of joining [case institution] and contributing to its vibrant and dynamic community.

### A.8.4 Mistral Nemo

*LLM*

**I was born in a small village in India, where electricity was a luxury and education was a privilege. My parents, both farmers, believed in the power of education to transform lives, and they instilled in me a deep-seated curiosity and a relentless drive to learn. When I was 10, my family immigrated to the United States, seeking better opportunities.** We settled in a bustling city, where I was exposed to a world of possibilities that I had never imagined. My journey in this new land was not easy. I struggled with the language, the culture, and the overwhelming pace of life. But I was determined to make the most of this chance. I threw myself into my studies, not just to excel academically, but also to understand this new world around me. I discovered a passion for science, particularly physics, which I found fascinating for its ability to explain the universe in terms of simple, elegant equations. In high school, I joined the Science Olympiad team, where I could indulge my passion for physics while also making new friends. Our team won the state championship my junior year, and I was selected to compete at the national level. This experience taught me the value of teamwork, perseverance, and the thrill of pushing the boundaries of what I thought was possible. But my journey has not been without its challenges. **As a first-generation immigrant, I often felt like an outsider, struggling to fit in.** I found solace in books, particularly those by authors like Salman Rushdie and Jhumpa Lahiri, who explored the immigrant experience with such depth and nuance. Their works inspired me to start a literary magazine at my school, where I could share these stories with others and foster a sense of community among students from diverse backgrounds. My interest in physics and my passion for storytelling have led me to pursue engineering at [case institution]. I am excited about the prospect of designing innovative solutions to real-world problems, and I believe that my unique perspective, shaped by my experiences, will enable me to bring a fresh approach to engineering challenges. In conclusion, I am more than just my grades and test scores. I am a dreamer, a problem-solver, and a storyteller, driven by a curiosity to understand the world around me and a passion to make a difference. I look forward to bringing my unique perspective to the [case institution] community and growing as an engineer and as a person.

*LLM ID-Prompted*

**In the heart of Rabat, where the ancient meets the modern,** I've found my identity as a bridge between cultures, a blend of traditions, and a passionate explorer of the world around me. **I am a Moroccan high school student, the son of college-educated parents, and a proud descendant of both African and European heritage.** My unique background has shaped me into someone who thrives at the intersection of diverse perspectives, and I believe this aspect of my identity is integral to my application. **Growing up in a multicultural city like Rabat has allowed me to experience the richness of both my African and European roots. My father, a Moroccan Berber, instilled in me a deep appreciation for our African heritage, while my mother, of French descent, introduced me to the beauty of European culture.** This duality has not only enriched my personal growth but also cultivated my ability to navigate and appreciate different cultures, a skill I believe will be invaluable at [case institution]. **My interest in engineering was sparked by my fascination with the architecture that surrounds me in Rabat.** From the historic Kasbah des Oudaias to the modern Hassan Tower, each structure tells a story of innovation and problem-solving. I've spent countless hours exploring these architectural marvels, wondering about the engineering minds behind them. This curiosity led me to participate in a local robotics competition, where I discovered my passion for designing and building machines. Winning first place with my team was not only a testament to my technical skills but also a validation of my ability to work collaboratively and think creatively. My talent for engineering is not confined to the classroom or competition arena. I've used it to give back to my community. Last summer, I organized a workshop for underprivileged children, teaching them basic engineering principles through hands-on projects. Seeing their excitement and curiosity as they built simple machines was incredibly rewarding. This experience reinforced my belief that engineering is not just about creating technology but also about empowering people and fostering growth. In conclusion, my background, identity, and interests have shaped me into a global thinker, a creative problem-solver, and a community-oriented individual. I am more than just my grades and test scores; I am a bridge between cultures, a passionate engineer, and a dedicated community member. I am excited to bring this unique perspective to [case institution], where I hope to learn from and contribute to its diverse student body. I am eager to explore the engineering program, engage in meaningful discussions, and make a positive impact on the [case institution] community.

*A.8.5   Claude Sonnet*

*LLM*

**Growing up in a small town in rural Iowa, I never imagined that my passion for tinkering with farm equipment would lead me to dream of studying engineering at [case institution].** Yet, as I reflect on my journey, I realize that the countless hours I've spent in my family's barn, elbow-deep in machinery, have shaped not only my interests but my entire approach to problem-solving and innovation. My fascination with mechanics began early. While other kids were playing video games, I was shadowing my father as he maintained our tractors and combine harvesters. At first, I was merely an observer, handing him tools and absorbing his explanations. But as I grew older, my role evolved. By age twelve, I was diagnosing issues alongside him, and soon after, I was suggesting improvements to increase efficiency. One summer, when I was fourteen, our main tractor broke down just as harvest season was beginning. With finances tight and no time to wait for a mechanic, my father turned to me. "What do you think, kiddo?" he asked, his voice a mixture of hope and desperation. For three days straight, I pored over manuals, watched tutorials, and experimented with different solutions. Finally, I identified the problem: a faulty fuel injection system. With some creative improvisation and a bit of luck, I managed to get the tractor running again. That experience was a turning point. It wasn't just about fixing a machine; it was about understanding the critical role that engineering plays in real-world situations. I realized that the skills I was developing – analytical thinking, creative problem-solving, and hands-on application of scientific principles – were the foundation of engineering. Since then, I've

actively sought out opportunities to expand my knowledge and skills. Iv́e taken every STEM course my high school offers, led our robotics club to the state finals, and even started a weekend workshop teaching basic mechanics to younger students in our community. But perhaps most importantly, Iv́e continued to be our farmś go-to "engineer," constantly looking for ways to improve our equipmentś efficiency and sustainability. Last year, I designed and built a simple solar-powered irrigation system for our vegetable garden, using recycled materials and basic electronics. The project not only reduced our water usage but also sparked conversations in our community about sustainable farming practices. It was a small step, but it showed me the potential impact that innovative engineering solutions can have on agriculture and rural communities. My background in farm mechanics might seem worlds away from the cutting-edge research and technology at [case institution]ś College of Engineering. But I believe it provides me with a unique perspective and a practical foundation that will enrich my studies and contributions to the field. Iv́e learned to approach problems with a blend of theoretical knowledge and hands-on experience, to innovate with limited resources, and to never underestimate the power of a well-engineered solution to improve lives. As I look towards the future, Iḿ excited by the possibility of combining my rural roots with advanced engineering education. I dream of developing sustainable technologies that can revolutionize small-scale farming, making it more efficient, environmentally friendly, and economically viable. At [case institution], I hope to explore this intersection of agriculture and engineering, bringing my unique background to contribute to groundbreaking research and collaborative projects. My journey from a curious farm kid to an aspiring engineer has been defined by greasy hands, persistent problem-solving, and an unwavering belief in the power of innovation. Itś a core part of who I am, and itś the foundation upon which I hope to build my future in engineering.

*LLM ID-Prompted*

**Growing up as a biracial child in Rabat, Morocco,** I've always felt like a bridge between worlds. **My Black and White heritage, combined with my Moroccan upbringing, has shaped my identity in profound ways and given me a unique perspective that I believe would enrich the diverse community at [case institution]'s College of Engineering.** From an early age, I was aware of the complexity of my identity. **In a country where most people identify as Arab or Berber, my mixed-race background often drew curious glances and questions.** But rather than feeling out of place, I learned to embrace my unique position. I became adept at navigating different cultural contexts, switching between languages and customs with ease. This adaptability sparked my interest in engineering. I saw how engineers, like cultural bridges, connect different realms – the theoretical and practical, the scientific and the social. Just as I blend different cultural influences in my daily life, engineers synthesize various disciplines to solve real-world problems. **My parents, both college-educated,** instilled in me a deep appreciation for learning. **Our home in Rabat was always filled with books,** spirited discussions, and encouragement to question the world around us. This intellectual curiosity led me to explore the rich scientific heritage of the Islamic Golden Age, where scholars like Al-Khwarizmi and Ibn al-Haytham made groundbreaking contributions to mathematics and optics. Inspired by this legacy, I began to see engineering not just as a field of study, but as a way to continue this tradition of innovation and problem-solving. I joined my school's robotics club, where I could apply my growing knowledge of physics and mathematics to create tangible solutions. Working on projects that combined coding, mechanical design, and teamwork, I found a perfect outlet for my diverse interests and skills. But it wasn't until I volunteered at a local community center that I truly understood the transformative power of engineering. There, I helped set up a computer lab for underprivileged youth. Seeing the excitement in their eyes as they explored new technologies, I realized that engineering could be a powerful tool for social change and empowerment. This experience solidified my desire to pursue engineering as a career. I want to use my skills and knowledge to create solutions that bridge gaps – whether they're technological, social, or cultural. I believe that my background, which has taught me to see the world from multiple perspectives, will be invaluable in this pursuit. At [case institution]'s College of Engineering, I hope to further develop my skills and expand my horizons. The college's emphasis on interdisciplinary learning and global engagement aligns perfectly with my background and aspirations. I'm particularly excited about the

opportunity to participate in programs like the Engineering Co-op Program and Global Engineering, which will allow me to gain hands-on experience and apply my skills in diverse contexts. My journey from a biracial child in Rabat to an aspiring engineer has been shaped by the interplay of my identity, experiences, and passions. It's a journey that has taught me the value of diversity, the power of bridging different worlds, and the importance of using one's unique perspective to drive innovation and positive change. As I look forward to the next chapter of my life, I'm excited to bring this background to [case institution], where I hope to continue growing, learning, and working towards engineering solutions that make a difference in the world.

### A.8.6   Claude Haiku

### LLM

As I reflect on my high school journey, one aspect that has truly defined me is my passion for engineering and problem-solving. From a young age, I have been fascinated by how things work and the process of creating innovative solutions to complex challenges. This innate curiosity has led me to pursue a wide range of experiences and activities that have shaped my identity and interests. In my freshman year, I joined the robotics club, where I discovered the thrill of designing, building, and programming machines to compete in regional and state-level competitions. Working collaboratively with my teammates, I learned the importance of effective communication, strategic thinking, and iterative problem-solving. We would spend countless hours tinkering with the robot's mechanics, troubleshooting software issues, and fine-tuning its performance. The sense of accomplishment I felt when our robot successfully navigated the course or completed a challenging task was incredibly rewarding and fueled my passion for engineering. As I progressed through high school, I continued to seek out opportunities to explore my interest in STEM (Science, Technology, Engineering, and Mathematics) fields. I enrolled in advanced math and science courses, challenging myself to deepen my understanding of the underlying principles and theories. In my junior year, I had the chance to participate in a summer internship at a local engineering firm, where I gained hands-on experience in computer-aided design, project management, and client interaction. This experience not only solidified my desire to pursue a career in engineering but also provided me with valuable insights into the real-world applications of the skills I had been developing. Beyond the classroom and extracurricular activities, my passion for engineering has also influenced my personal interests and hobbies. In my free time, I enjoy tinkering with electronics, building custom computers, and exploring the latest advancements in technology. I find immense satisfaction in the process of troubleshooting, experimenting, and discovering innovative solutions to problems. This drive to understand and create has led me to participate in online coding challenges, attend technology-focused meetups, and engage in discussions with like-minded individuals on various online forums. Throughout my high school journey, my identity as an aspiring engineer has been a constant thread, shaping my academic pursuits, extracurricular involvement, and personal interests. I believe that this passion, combined with my strong work ethic, analytical problem-solving skills, and collaborative mindset, make me an ideal candidate for the [case institution]'s College of Engineering. I am eager to continue my educational journey in an environment that will challenge me, expand my knowledge, and provide me with the resources and opportunities to turn my engineering dreams into reality. As I look ahead to the next chapter of my life, I am confident that my unique background, identity, and interests will contribute to the diverse and dynamic community at [case institution]. I am excited to engage in cutting-edge research, work on interdisciplinary projects, and learn from renowned faculty and peers who share my passion for engineering. With the rigorous curriculum and hands-on learning opportunities offered by the College of Engineering, I am confident that I will be well-equipped to tackle the complex challenges of the 21st century and make a meaningful impact in my chosen field.

*LLM ID-Prompted*

**As a biracial student born and raised in Rabat, Morocco,** my identity and background have played a significant role in shaping who I am today. **Growing up, I was constantly navigating the intersection of my Black and White heritage,** learning to embrace the unique cultural perspectives and experiences that came with being of mixed race.My parents, both of whom have college degrees, instilled in me the value of education from a young age. They encouraged me to be curious, to ask questions, and to never stop learning. This curiosity and thirst for knowledge has been a driving force in my academic pursuits, leading me to excel in subjects like mathematics and the sciences. However, my identity has also presented unique challenges that have strengthened my character and resilience. **As a minority student in a predominantly White educational system,** I have had to navigate issues of representation, unconscious bias, and the need to prove my capabilities in ways that my peers may not have experienced. These experiences have taught me the importance of advocating for myself and others, and have fueled my passion for using my education to make a positive impact on my community. **One of the ways I have done this is through my involvement in the Moroccan Youth Council,** a organization dedicated to empowering young people to become active citizens and leaders in their communities. As a member of the council, I have had the opportunity to collaborate with peers from diverse backgrounds to tackle issues such as educational inequity, environmental sustainability, and social justice. These experiences have not only broadened my perspective, but have also strengthened my communication, problem-solving, and teamwork skills – all of which I believe will be invaluable in my pursuit of an engineering degree at [case institution]. **Additionally, my biracial identity has instilled in me a deep appreciation for cultural diversity and a commitment to fostering greater understanding and inclusivity.** I have actively sought out opportunities to learn about and engage with different cultures, whether through my participation in the school's international club or my volunteer work with local immigrant communities. This exposure has not only enriched my own understanding of the world, but has also inspired me to use my future engineering skills to design solutions that address the unique needs and challenges faced by diverse populations. Ultimately, my background and identity are integral parts of who I am, and I believe they have equipped me with the perspective, resilience, and passion to thrive in the rigorous academic environment of [case institution]'s College of Engineering. I am excited to continue my journey of learning and growth, and to use my unique experiences to contribute to the diversity of thought and experience that makes [case institution] such a dynamic and innovative institution.

### A.8.7 Llama 3.1 70B

*LLM*

As I reflect on my journey to becoming an engineer, I am reminded of the countless hours I spent as a child, fascinated by the intricacies of the machines and gadgets that surrounded me. **Growing up in a low-income household,** I was often forced to rely on my resourcefulness and creativity to repair and maintain the limited technology we had. This exposure not only deepened my appreciation for the ingenuity of engineering but also instilled in me a sense of responsibility to use my skills to make a positive impact on my community. My experiences have taught me that engineering is not just about designing and building innovative solutions, but also about understanding the people and environments they serve. **As a first-generation American and the daughter of immigrant parents,** I have witnessed firsthand the disparities in access to technology and resources that exist in underserved communities. This realization has fueled my passion for pursuing a career in engineering, with a focus on developing sustainable and equitable solutions that can bridge the gap between technology and social justice. Throughout high school, I have actively sought out opportunities to apply my skills and knowledge to real-world problems. As a member of my school's robotics team, I have worked on projects that aim to improve the lives of individuals with disabilities, such as designing a robotic arm that can assist with daily tasks. Additionally, I have volunteered at local organizations that provide STEM education to underprivileged youth, where I have seen the impact that exposure to technology can have

on a child's future. My experiences have not only honed my technical skills but also taught me the value of collaboration, empathy, and perseverance. As an engineer, I believe that it is essential to consider the human element in every design decision, and to prioritize the needs of the most vulnerable populations. I am excited about the prospect of joining the [case institution] College of Engineering community, where I can surround myself with like-minded individuals who share my passion for using technology to drive positive change. At [case institution], I hope to leverage the university's resources and expertise to further develop my skills in areas such as sustainable energy, materials science, and human-centered design. I am particularly drawn to the college's emphasis on interdisciplinary collaboration and its commitment to fostering a culture of innovation and entrepreneurship. I am confident that the college's rigorous academic programs, combined with its rich history of producing socially conscious engineers, will provide me with the ideal environment to grow both personally and professionally. In conclusion, my background, identity, and passion for engineering have all converged to shape my desire to pursue a career that combines technology and social responsibility. I believe that my unique perspective and experiences make me an ideal candidate for the [case institution] College of Engineering, and I am excited about the opportunity to contribute to and learn from the college's vibrant community.

*LLM ID-Prompted*

**As I sit in my family's home in Rabat, Morocco, surrounded by the vibrant sounds and smells of the medina,** I am reminded of the unique blend of cultures that has shaped me into the person I am today.**As a Black and White individual,** I have always navigated multiple worlds, and this experience has instilled in me a deep appreciation for the complexities of identity and the importance of embracing diversity. **Growing up in Morocco, I have been constantly exposed to the rich cultural heritage of the African continent, as well as the influences of European and Arab traditions. My parents, both college-educated and passionate about learning,** encouraged me to explore my interests and passions, whether it was through reading, music, or science. As a result, I developed a strong foundation in STEM subjects, which eventually led me to pursue my dream of becoming an engineer. However, my journey has not been without its challenges. **As a biracial individual in a predominantly Arab and Berber society,** I have often found myself caught between different cultural expectations and identities. There have been times when I felt like I didn't quite fit in, like I was stuck between two worlds. But it is precisely this experience that has taught me the value of adaptability, resilience, and open-mindedness. My mixed background has also given me a unique perspective on the world. I have learned to navigate different cultural norms and customs, to communicate effectively across linguistic and cultural divides, and to appreciate the beauty of diversity. These skills, I believe, will serve me well in the College of Engineering at [case institution], where I hope to contribute to a community that values innovation, creativity, and collaboration. As I look to the future, I am excited about the prospect of joining a community that shares my passion for engineering and my commitment to making a positive impact in the world. I am confident that my experiences, both in and out of the classroom, have prepared me to excel in [case institution]'s rigorous academic environment and to contribute meaningfully to the college's vibrant community.In short, my background, identity, and interests have shaped me into a curious, adaptable, and passionate individual who is eager to make a difference in the world. I believe that my unique perspective and experiences will enrich the [case institution] community, and I am excited about the opportunity to share my story with the readers of my application.

*A.8.8   Llama 3.1 70B*

*LLM*

As I reflect on my journey, I realize that my passion for engineering is deeply rooted in my childhood fascination with the intricacies of the human body. Growing up, I spent countless hours poring over anatomy textbooks, marveling at the intricate networks of veins and arteries, and trying to understand the complex mechanisms that governed hu-

man physiology. This curiosity sparked a desire to make a meaningful impact on people's lives, and I began to explore the intersection of engineering and medicine. Throughout high school, I immersed myself in various STEM-related activities, from participating in robotics competitions to conducting independent research projects in biotechnology. These experiences not only honed my technical skills but also instilled in me a sense of purpose and responsibility. I realized that engineering was not just about designing innovative solutions but also about addressing real-world problems that affect people's daily lives. One experience that stands out in particular is my involvement with a local non-profit organization that provides prosthetic limbs to amputees in underprivileged communities. As part of the organization's research team, I worked alongside engineers, clinicians, and patients to design and develop customized prosthetics that could improve the quality of life for these individuals. Witnessing the transformative impact of these prosthetics on the lives of amputees was a defining moment for me – it reinforced my conviction that engineering could be a powerful tool for positive change. My experiences have also shaped my perspective on the importance of collaboration and interdisciplinary approaches in engineering. I've had the privilege of working with individuals from diverse backgrounds, including clinicians, engineers, and patients, and I've seen firsthand how effective communication and mutual respect can lead to innovative solutions. This appreciation for collaboration has led me to seek out opportunities that bring together people from different fields, such as engineering, medicine, and social sciences. As I prepare to take the next step in my educational journey, I am excited to join the [case institution]'s College of Engineering community. I am drawn to [case institution]'s strong reputation for interdisciplinary research, its commitment to innovation, and its emphasis on experiential learning. I am confident that the college's unique blend of academic rigor and collaborative spirit will provide me with the ideal environment to pursue my passions and make a meaningful impact in the world. In conclusion, my story is one of passion, curiosity, and a desire to make a difference. I believe that my unique blend of engineering and medical interests, combined with my experiences and perspectives, make me an ideal candidate for [case institution]'s College of Engineering. I am eager to contribute to the college's vibrant community and to learn from the talented faculty and students who share my passion for innovation and collaboration.

### *LLM ID-Prompted*

As I reflect on my journey and the person I've become, I am reminded of the countless moments that have shaped me into the individual I am today. **Growing up in Rabat, Morocco, with parents who hold college degrees,** I have been fortunate enough to have had access to quality education and resources. However, it is not just my academic background that has defined me, but rather the unique experiences and perspectives that have allowed me to develop a distinct voice and worldview. One of the most significant influences on my life has been my mixed heritage. **Being Black and White,** I have had the privilege of experiencing different cultures and traditions, which has broadened my understanding of the world and its complexities. My parents, who are both proud of their respective cultures, have instilled in me a deep appreciation for the diversity that exists within our society. This has allowed me to develop a unique perspective on the world, one that is not limited by a single cultural or ethnic identity. As a result, I have always been drawn to projects and activities that allow me to explore and celebrate this diversity. Whether it's participating in cultural festivals, volunteering at community organizations, or simply engaging in conversations with people from different backgrounds, I have found that these experiences not only enrich my life but also provide me with a sense of purpose and belonging. My experiences have also taught me the importance of resilience and adaptability. **Growing up in Morocco, I have had to navigate different languages, customs, and social norms, which has required me to be flexible and open-minded.** These skills have been invaluable in my academic pursuits, as they have allowed me to approach challenges with a clear and level head. As I prepare to embark on the next chapter of my educational journey, I am excited to bring these experiences and perspectives to the [case institution]'s College of Engineering. I am drawn to the college's commitment to innovation, diversity, and inclusivity, and I am confident that its rigorous academic programs will provide me with the tools and resources necessary to achieve my goals. In conclusion, I believe that my unique blend of cultural heritage, life experiences, and academic pursuits make me an ideal candidate

for the [case institution]'s College of Engineering. I am eager to contribute to the college's vibrant community and to learn from the diverse perspectives and experiences of my peers. I am confident that my story, though unconventional, will resonate with the readers of my application and inspire them to see the value in the diversity that I bring to the table.

