# OpenReview forum: "Poor Alignment and Steerability of Large Language Models: Evidence Using 30,000 College Admissions Essays"
_colmweb.org/COLM/2025/Workshop/Social_Sim — Social Sim'25_

### Official Review · Reviewer_hr3j · 2025-07-04
**Limited methodological novelty despite a timely and relevant study on LLM alignment in admissions essays**

**Rating:** 6
**Overall Assessment:** 3
**Confidence:** 3

**Review:**

This paper addresses the alignment and steerability of LLMs in a high-stakes admissions essay setting, leveraging a large, real-world dataset. The topic is timely and relevant, and the empirical findings are clearly presented. However, the methodological contribution is limited, relying on simple vector-space analyses without deeper modeling, and the private dataset hinders reproducibility. Strengthening the theoretical insights and improving the experimental rigor would enhance its significance.

**Comments Suggestions And Typos:**

The manuscript is clearly written and well organized; I have no substantive comments or typos to report.

**Paper Summary:**

This paper investigates the alignment and steerability of LLMs in the high-stakes context of college admissions essays. Using a dataset of 30,000 human-written essays, the authors compare them against LLM-generated texts created with either a generic prompt or a prompt including the human author's specific demographic information. The study consistently finds that LLM-generated texts are linguistically distinct from human essays, and that identity-specific prompting is remarkably ineffective at steering the models to align with the writing patterns of target demographic groups. Furthermore, the analysis reveals that prompting does not alleviate homogenization, as synthetic essays remain more similar to each other than to human writing. These findings demonstrate the poor alignment and steerability of current LLMs, raising serious concerns about their use in applications requiring authentic, individualized expression.

**Relevance:**

5

**Summary Of Strengths:**

1. Unique real-world dataset: Utilizes a large corpus (~30 k) of authentic admissions essays, offering rare, high-stakes context for studying LLM alignment and steerability.

2. Comprehensive model coverage: Evaluates eight diverse LLMs (GPT-4o, Claude 3, Llama-3, Mistral, etc.), providing robust cross-model evidence and enhancing result generality.

3. Clear experimental reporting: Presents pipeline, metrics, and limitations transparently, with well-organized figures/tables that make the empirical findings easy to follow.

**Summary Of Weaknesses:**

1. Methodological depth is limited: The study relies mainly on basic vector-space similarity and logistic regression, without deeper modeling or causal analysis to explain why steerability fails.

2. Reproducibility concerns: The private, non-shareable admissions-essay corpus (and scant IRB/ethical details) prevents independent replication and verification of the results.

---

### Official Review · Reviewer_AKhw · 2025-07-14
**the generation style of LLM is not aligned with human given demographic info**

**Rating:** 5
**Overall Assessment:** 2
**Confidence:** 4

**Review:**

In general, the paper presents a series of experiments with a very large amount of data that are valuable for the community to be aware of. This would be the reason for why it may deserve to be published.

The problem of this paper is that their experiments seems to be very weak to connect back to the core motivation of the paper. The first concern is about the prompt method employed (just a few sentences of demographic info, no In-context learning). The second concern is about real world usage of LLM (compare entire article generation instead of rephrasing). I'll elaborate in the strength and weakness section.

One other major concern is that it might not fit our workshop theme well? Their simulation doesn't relate to social *interaction*  at all.  My understanding of this workshop is either to do social interaction simulation to investigate some topic, or to develop methods to help do better social interaction?

Therefore if the AC believes this paper is relevant, my opinion is accept, otherwise would be no.

**Comments Suggestions And Typos:**

n/a

**Ethical Concerns:**

The data are actual student's college application essays. So it's definitely related to privacy issues. But they did not release it so it should be fine? I am not clear of if there's any standard of practice in this scenario.

**Paper Summary:**

This paper is motivated by whether the use of LLM to help writing would make the uniqueness of student's writing style disappear.  Therefore they compared the writing style of LLM generated text with human generated, with and without group identity prompt (age, sex,ethnicity, demographics etc). They found LLM generated texts are more similar to each other with or without identity prompt than with the human. Therefore it shows, at least based on identity prompt, LLM can't be steered to generate according to these group's style.

**Relevance:**

2

**Summary Of Strengths:**

The experiments are conducted on a extensive amount of data. They use 29232 essays from actual college student applicants, and LLMs generate double amount (with / without ID prompt). Large scale experiments are always valuable.

**Summary Of Weaknesses:**

The motivation is of this paper is whether the use of LLM to help writing would make the uniqueness of student's writing style disappear.
Although the experiments are designed to be correlated to this motivation by checking how LLM generation can be steered to be similar to the diverse groups, the way of steering doesn't seem to be convincing for that big motivation.
1. the prompt method just provides the sex, ethnicity, demographics and first-generation status, it would be hard for human to make sense of what that "style" should be. For such task, in-context learning (with a few examples of that group) should be considered
2. the evaluation is to have LLMs to generate the entire article to compare with human's articles. However, in real sceneario when people use LLMs for writing, often we use it for paragraph rephrasing instead of the whole paper generation. This is a very different task.

---

### Meta-Review · Program_Chairs · 2025-07-24

**Recommendation:** Accept

**Metareview:**

Please incorporate the feedback!